# Effect of Different Post-Harvest Processing Methods on the Chemical Constituents of *Notopterygium franchetii* by an UHPLC-QTOF-MS-MS Metabolomics Approach

**DOI:** 10.3390/molecules24173188

**Published:** 2019-09-02

**Authors:** Xueyan Su, Youjiao Wu, Ying Li, Yanfei Huang, Yuan Liu, Pei Luo, Zhifeng Zhang

**Affiliations:** 1Institute of Qinghai-Tibetan Plateau, Southwest Minzu University, Chengdu 610041, Sichuan Province, China; 2State Key Laboratories for Quality Research in Chinese Medicines, Faculty of Chinese Medicine, Macau University of Science and Technology, Macau 853, China

**Keywords:** *Notopterygium franchetii*, post-harvest processing, UHPLC-DAD-QTOF-MSMS, chemical constituents

## Abstract

*Notopterygium franchetii* is a herb used in traditional Chinese medicine, where it is known as *qianghuo*. Its bioactive qualities are influenced by the post-harvest processing methods used (such as drying). However, changes in chemical components according to the drying method are unknown. Fresh roots and rhizomes of *N. franchetii* were subjected to seven drying methods. Chromatography-mass spectrometry combined with targeted and untargeted analyses were used to investigate relationships between drying methods and chemical concentrations. According to targeted evaluations of the six main bioactive constituents, their total contents decreased significantly in all drying methods. Hierarchical clustering analysis of the drying methods and total metabolome detected 30 chemical constituents, for which heap maps were obtained. Hot air drying was the best processing method, producing the least chemical changes at the lowest cost, while shade drying caused the greatest chemical changes. In conclusion, the wide range of chemical changes in *N. franchetii* caused by drying was investigated. Such changes potentially affect the quality of herbal medicines.

## 1. Introduction

Drying is the most fundamental procedure in the post-harvest processing of Chinese herbal medicines, as it affects their bioactive constituents and economic value [1]. Previous studies have shown that the drying of medicinal plants comprises an intense drought stress response. Plant secondary metabolites are usually produced under environmental stresses such as low water and high temperature [2]. Generally, it is believed that the concentration of bioactive components accumulates in the pre-harvest period and decreases during the post-harvest drying process, which is of long duration and uses high temperatures [3]. Accordingly, research into Chinese medicine drying methods aims to optimize the contents and types of bioactive constituents existing in the plant. 

*Notopterygium franchetii* (NF), family Umbelliferae, has been used as a traditional Chinese medicine for thousands of years. It is officially named *qianghuo* in the Chinese Pharmacopeia (2015 edition). Until now, NF has mainly been cultivated in high altitude regions (2500–3000 m) in the Sichuan and Qinghai provinces of China. Its rhizomes and roots are widely used in Chinese medicine to treat conditions such as rheumatoid arthritis, common fever, subcalorism, and headache [4]. 

Phytochemical studies have shown that coumarins, phenoloids, flavonoids and their glycosides, and essential oils are the main types of chemicals in NF [5,6,7]. Among these, coumarins, especially isoimperatorin, notopterol, and bergapten, are well known to be the major bioactive compounds, and contribute to NF’s anti-inflammatory, analgesic, and anti-cancer activities [8,9,10]. These coumarins were also found to increase oxidative stress and HO-1 protein levels in human fetal hepatocytes [11]. In the Chinese Pharmacopoeia, the total content of isoimperatorin and notopterol (>0.4%, g/g) was used to evaluate the quality of commercial NF. However, previous studies found that different batches of NF had significantly different contents of these two components. The range of isoimperatorin and notopterol present in 28 samples collected from herbal markets was 0.6–3.7% (g/g) [12,13,14]. It was found that the content of both isoimperatorin and notopterol was significantly different in all the samples, which has implications for the clinical efficacy of NF. These variations were commonly ascribed to the different origins, collection times, growing environments, climatic factors, and soil microorganisms of the NF batches. However, another very important factor is the post-harvest processing method, which can significantly affect the quality and economic value of medicinal plants [15,16]. 

In our previous pilot experiments, post-harvest drying was observed to cause changes in the contents of coumarins, phenoloids, and essential oils. Thus, NF may be unstable in response to temperature. Traditionally, sunlit or ventilated shade drying is commonly used in post-harvest processing [17]. However, due to their low efficiency and long duration, these two drying methods have become restricted in the modern Chinese medicine industry. Furthermore, mustiness or bacteria often cause medicinal deterioration after long drying durations [18]. In recent years, sulfur fumigation has frequently been employed during post-harvest drying to prevent mustiness. However, increased SO_2_ levels and heavy metal contamination can cause deterioration that seriously degrades the quality of herbal medicines [19]. Therefore, in recent years, some safer and more effective post-harvest drying methods have been developed. These include the use of far-infrared radiation, microwaves, vacuums, and hot air [20]. Currently, there has been little research on the effects of these modern drying techniques on the quality of NF.

Several analytical methods have been developed for the evaluation of NF, such as high-performance liquid chromatography (HPLC) and gas chromatography (GC) [21,22]. However, GC technology is not suitable for the analysis of thermo-labile components such as isoimperatorin and notopterol. Existing studies have only used HPLC or HPLC-MS to quantify a few compounds according to the origin of NF samples [23]. There are no reports on changes to the chemical constituents of NF arising from different drying methods. 

In the present study, an ultra-high-performance liquid chromatography photodiode array detector was combined with quadrupole time-of-flight mass spectrometry (UHPLC-DAD-QTOF-MS/MS) to analyze the chemical components of NF quantitatively and qualitatively. The focus was on the quality-related chemical changes occurring in NF due to different drying methods. The same batch of NF was used to create triplicate samples that were subjected to sun drying, shade drying, microwave drying, freeze drying, oven drying, and infrared drying. Different chemical constituents were identified through the molecular ions and fragment ions of the QTOF-MSMS analysis and those reported in the literature, and through the determination of standard compounds. Statistical analysis was performed to identify the optimal post-harvest process.

## 2. Results and Discussion 

### 2.1. Method Validation and Optimization

#### 2.1.1. Sample Preparation Optimization

First of all, the different drying methods were performed and optimized. The results demonstrated that traditional sunlight or shade drying were lower in efficiency and longer in duration than modern drying technologies such as microwave drying, freeze drying, and infrared radiation drying. Hot air drying and vacuum drying are usually used in modern industry. Most remarkably, microwave radiation produced a thermal effect that penetrated the material to speed up dehydration, making it the fastest drying method. 

The extraction methods were also optimized according to their methods, solvents, and durations. Two commonly-used extraction methods were investigated: hot reflux and ultrasonic-assisted extraction. Determination of the extraction efficiency of NF was performed by UHPLC. In the *Chinese Pharmacopeia*, notopterol and isoimperatorin as the major bioactive constituents of NF, were used as chemical markers to evaluate NF quality. The results indicate that ultrasonic-assisted extraction was most suitable as it provides the maximum peak area and best resolution. Additionally, extraction solvents (100%, 70% or 50% methanol) and durations (10, 20, 40, and 60 min) were also screened. The combination of 70% methanol and 40 min was selected because it demonstrated superior extraction capacity.

#### 2.1.2. Chromatography and MS Parameter Optimization

In order to improve chromatographic separation, a Waters Acquity HSS T_3_ (100 mm × 2.1 mm, 1.8 m) column and a Waters Acquity BEH C_18_ (100 mm × 2.1 mm, 1.7 m) column were investigated for option of chromatography columns. Column temperatures of 25, 30, 35, and 40 °C) were compared. The results showed that the Waters Acquity HSS T_3_ column achieved the best baseline separation for most sample constituents at 30 °C. A mobile phase composed of two organic phases (MeOH, acetonitrile) and an aqueous phase (water, formic acid) were tested. It was found that MeOH and acetonitrile presented different elution abilities, with acetonitrile giving better for peak separation. Furthermore, the aqueous phase containing formic acid not only improved the resolution of the chromatographic peaks but also produced the adduct [M − H + HCOOH]^−^, which was beneficial for verifying the molecular ions [M − H]^−^. Therefore, an acetonitrile–water and 0.1% formic acid mixed mobile phase system was finally selected to achieve the best separation. To improve the sensitivity for the six analytes, their wavelengths were also optimized. Considering the maximum ultraviolet absorptions corresponding to the analytes, 330 nm was selected as the determination wavelength for all components.

Mass analysis was acquired in both positive and negative ion modes. The mass-to-charge ratio (*m*/*z*) was set within the range 100–1700 *m*/*z*. The negative ion model was chosen due to its better signal-to-noise ratio and greater amount of peak information. In order to generate more precursor ions and fragments ions. Under the optimized chromatographic and MS parameters, 30 components of the NF samples could be simultaneously detected in one run within 40 min. A representative base peak ion chromatogram analyzed in the negative mode is shown in Figure 1.

### 2.2. Methodological Validation of UHPLC

The contents of nodakenin (**8**), psoralen (**10**), bergapten (**20**), notopterol (**23**), imperatorin (**28**), and isoimperatorin (**29**) in NF were determined by UHPLC. All produced good linearities (*r*^2^ > 0.9992) within the acquired ranges. The LODs and LOQs of the six analytes were 13–40 ng and 26–78 ng, respectively, suggesting that the method was sensitive enough to determine the six analytes. The relative standard deviation (RSD) of the intra-day and inter-day precisions of the six analytes were less than 1.12%. The RSD of repeatability and storage stability were less than 2.03% and 2.87%, respectively. The recovery was 96.52–103.48% (RSD < 2.56%). These results suggested that the developed method was able to accurately determine the six analytes accurately (Table 1).

### 2.3. Quantification of Six Analytes in Fresh NF and NF Dried by Different Methods

Six bioactive components in NF samples dried by different methods were quantified. Identification of these six analytes was further confirmed by comparing the retention times, UV spectra and accurate masses of molecular ions or fragment ions with those of reference compounds. The results are shown in Figure 1 and Table 2. It was found that nodakenin (**8**) and isoimperatorin (**29**) were the main components of all NF samples, and the determination results of the six analytes showed significant differences (*p* < 0.05) in most samples dried by different methods. Compared with fresh NF, the total contents of the six analytes in all dried samples decreased in the range of 31.55% ~ 63.44%. The total content after hot air drying was the greatest (decrease of 31.55%), followed by sunlight drying and freeze drying. The content in microwave drying samples decreased by 63.44%, which was the most significant change in all samples. This interesting phenomenon suggests that microwave drying can significantly destroys some of the chemical constituents due to molecular violent vibration. The far infrared radiation drying samples decreased by 58.04% due to rapid increasing temperature leading to molecular violent vibrations.

The content changes of the six components may have arisen from significant differences in drying duration, pressure and temperature, except microwave drying and far infrared radiation drying. The drying temperature required for vacuum and hot air drying was the same, while the drying duration of vacuum drying was much shorter than those of hot air drying, which lead to greater degradation of all six components. It was found that vacuum drying at 50 °C and 100 Pa also significantly decreased the total content of the six components (a reduction of 59.47%), which was easily degraded under high pressure and heat. However, it should be noted that the determination of the six components were statistically higher dried with hot air drying at 50 °C.

The common characteristic of sunlight drying and hot air drying was that the drying duration was relatively shorter than shading drying. Obviously, a lower temperature can better preserve the bioactive constituents. In summary, both shade drying and hot air drying was superior to other drying methods. However, due to the longer duration of shading drying, hot air drying in the oven was the optional method due to its higher efficiency. Hence, it is a potential alternative to the traditional sunlight and shade drying methods. Microwave drying and freeze-drying were also inferior to hot air drying for NF samples. Therefore, all seven methods induced significant degradation of the six components in the NF samples. However, compared with the other six methods, hot air drying was the best due to its lesser degradation of chemical constituents. 

### 2.4. Identification of Chemical Constituents of N. Franchetii

A total of 30 peaks were monitored in the NF samples by the UHPLC-QTOF-MSMS method. Of these, 26 compounds were unambiguously or tentatively identified for the first time. After adding formic acid to the mobile phase, not only were the chromatographic peaks not only improved but [M − H + HCOOH]^−^ ions were produced, which contributed to the confirmation of the molecular ion. [M + H]^+^ was considered to further confirm the molecular ion. Comparing the retention times and UV spectra of compounds **8**, **10**, **20**, **23**, **28**, and **29** with those of the commercial standards, these compounds were unambiguously identified as nodakenin (**8**), psoralen (**10**), bergapten (**20**), notopterol (**23**), imperatorin (**28**), and isoimperatorin (**29**), respectively. Tentative characterization of all other constituents was based on molecular ion and MS/MS fragmentation patterns and relevant literature [24,25,26,27,28,29,30,31,32,33,34,35]. The results are shown in Figure 2 and Table 3. Where applicable, some compounds previously identified from the same botanical family were also used for characterization. The attributions and relative amounts of all 30 compounds in the eight samples are summarized in Table 4.

### 2.5. Chemical Transformation of NF during Drying

The UHPLC-DAD chromatograph showed that the changes in the chemical constituents from the NF extracts that were mostly quantitative analysis. Qualitative comparison after drying was also very important for the profile of all the different drying method (Figure 2). QTOF-MSMS spectra were used to visualize all of the chemical changes by as it was found that the QTOF-MSMS profile showed the most significant differences. It was evident that compound **17** was more sensitive to shade drying than any other compound. The increase in compound **17** after shade drying was 10.0-fold, while compound **8** decreased 2.2-fold. Qualitative analysis showed that compounds **8** and **18** were predominant in all tested samples. Variations in the main compounds were presented in Figure 3, with some of these results may be arising from the longer drying duration of shade drying. The variations associated with microwave, freeze and far infrared radiation drying were less produced than those from sunlight and shade drying because the drying times were shorter, while the relative contents of compounds **26**, **27**, and **28** differed by up to 2 -fold after shade drying. Therefore, drying duration was the most important factor in the drying process. The optimal dehydration method is hot air drying due to its low cost and high efficacy. This method was selected for the standardization of NF post-harvest processing.

### 2.6. Multivariate Analyses

Principal component analysis (PCA) and hierarchical cluster analysis (HCA) were performed to analyze the data. The chemical phenotypes of the eight different types of samples were well discriminated. The accumulative contribution rate of the first four principal components was 95.36% (PC1 = 46.52%, PC2 = 80.30%, PC3 = 91.14%). As shown in Figure 3A, all triplicates from each type were clustered together and separated from the other sample clusters. It was found that in all the processed samples, the duration of shade drying was greater than that of the other drying methods, resulting in the variation in the shade-dried samples being the greatest. The same result was obtained by HCA analysis (Figure 3B).

The HCA of the peak area indicated that all the compounds were clustered in five categories. The distance in the category including compounds **18**, **8**, **25**, **17**, and **27** was greater than that of the others, indicating the greatest variation in all chemical constituents (Figure 3C). A heat map of the separation of the compounds attributed was made to illustrate the relative variations in the eight processed groups (Figure 4). In the heat map, each colored cell indicates a concentration, with colors ranging from red to blue representing relative intensities varying from high to low, respectively. The 30 chemical components were classified into five clusters: phenolic acids, organic acids and their esters, flavonoids and flavone glycosides, coumarin, sterols and others. The heat map shows the changes in the relative chemical compositions for each type, showing the differences that could be related to the different drying processes. To illustrate the conspicuous changes in compound concentrations in these five chemical categories, detailed descriptions for each type are made below.

*Coumarins*: Coumarins are the main bioactive constituents in NF, therefore, their variation may affect the quality of NF. From the heat map, isomers of imperatorin were found to be significantly changed, increasing 8.62- and 9.38-fold after sun drying and oven drying, respectively. Imperatorin increased 9-fold during shade drying. Nodakenin was found to be significantly decreased by about 60% in all processed samples.

*Phenolic acids*: Phenolic acids are shown in the upper part of the heat map, which indicates that they underwent the greatest variation in concentration in all drying methods. Chlorogenic acid was significantly increased, by 3.47- and 3.31-fold, after shade drying and microwave drying, respectively. In contrast, 3,5-dicaffeoylquinic acid was undetectable in all processed samples. The results suggest that phenolic acids are unstable during drying.

*Flavonoids and flavonoid glycosides*: Content of flavonoids and flavonoid glycosides were also changed by the processing methods. Compared with fresh samples, compounds **3**, **13**, and **21** had higher contents after processing. This is consistent with previous studies reporting that flavonoids and flavonoid glycoside concentrations can increase during storage. Compound **3** also exhibited a newly generated peak in the shade-dried samples.

*Other notable changes*: β-Sitosterol was also a newly generated peak found only in shade-dried samples. Compounds **28** and **29** are a pair of isomers that can interconvert during drying. In shade-dried samples, the content of compound **28** increased 8-fold, while that of compound **29** decreased significantly.

This research has shown that the post-harvest processing of herbal medicines is the most critical factor affecting their quality. The drying process mimics intense drought stress, which changes the plant’s secondary metabolites and causes changes in the bioactive components relevant to herbal medicine. During the drying process, such enzymatic reactions can occur even as the plant is transported from the field to the processing factory [36]. Hence, the properties and taste of herbal medicines can be influenced by the drying method used.

## 3. Materials and Methods

### 3.1. Materials and Reagents 

HPLC-MS grade acetonitrile was purchased from Sigma-Aldrich (St. Louis, MO, USA). Analytical grade acetic acid and formic acid were obtained from a local company (Chengdu, China). Ultra-pure water was obtained from a Milli-Q system (Millipore, St. Louis, MO, USA). Reference compounds of nodakenin (**8**), psoralen (**10**), bergapten (**20**), notopterol (**23**), imperatorin (**28**), and isoimperatorin (**29**) were purchased from Kangbang Biotechnology Company (Chengdu, China). Fresh NF samples (Batch No. 20180802) were collected from Jiuzhaigou County, Sichuan Province, an indigenous NF cultivation region, and authenticated by Prof. Hao Zhang (School of Pharmacy, Sichuan University, Chengdu, China). The moisture content of the fresh NF sample was about 60%. Voucher specimens were deposited at Southwest Minzu University, Chengdu, China.

### 3.2. Sample Preparation

#### 3.2.1. Preparation of Standard Solutions

Nodakenin (**8**), psoralen (**10**), bergapten (**20**), notopterol (**23**), imperatorin (**28**), and isoimperatorin (**29**) were accurately weighed with amber volumetric flasks and dissolved in methanol to obtain stock solutions of standard compounds, which were stored at −20 °C. Working solutions were prepared with methanol by serial dilution of the stock solutions.

#### 3.2.2. Sample Processing

The roots and rhizomes of fresh NF were sliced into 0.3–0.5 cm-thick sections. These sections were freely mixed; then, 100 g of sections were selected and subjected to each drying process. The final weight of the sections varied within 3–5% after being weighed twice, and the sections were stored for further analysis. Each drying method was performed in triplicate. Shade drying was carried out in a ventilated room. Sunlight drying, hot air drying, and vacuum drying were carried out in the sun, a ventilated oven, and a vacuum drying oven, respectively. Microwave drying and infrared radiation drying were conducted in a microwave oven and infrared drier, respectively. The different drying processes, their parameters, and the final weights of the dried NF samples are listed in Table 5. The dried samples were stored in a glass drier for further analysis.

#### 3.2.3. Sample Preparation

Dried sample was ground to powder and a 0.5 g aliquot of NF powder (40 mesh) was accurately weighed and extracted in an ultrasonic cleaner with 10.0 mL of methanol-H_2_O (70:30, *v*/*v*) for 30 min. The extracted solution was filtered with a 0.22 µm PTEE syringe filter before injection into the UHPLC. 

### 3.3. UHPLC-DAD-QTOF-MS/MS Analysis 

Chromatographic analysis was performed using a UHPLC system with an Agilent 1290 diode array detector (DAD; Agilent Technologies, Santa Clara, CA, USA). Separation was performed on an Acquity HSS T_3_ column (100 mm × 2.1 mm, 1.8 μm; Waters, Milford, MA, USA). The mobile phase A was 0.1% formic acid in deionized water (*v*/*v*), and phase B was acetonitrile of LC-MS grade. A linear gradient was optimized as follows: 0 min, 10% B; 10.0 min, 20% B; 20 min, 30% B; 30 min, 40% B; 35 min, 60% B; and 40 min, 90% B. The flow rate was 0.3 mL/min, the injection volume was 2 μL, and the column temperature was maintained at 30 °C. The DAD spectra were scanned in the 200–400 nm range and chromatograms were obtained at 330 nm for nodakenin (**8**), psoralen (**10**), bergapten (**20**), notopterol (**23**), imperatorin (**28**), and isoimperatorin (**29**). A 2 μL injection volume was used for the standard and sample solutions.

Mass spectrometry was performed using a 6550 UHD Accurate-Mass Q-TOF/MS detector with an electrospray ion source (ESI). The negative mode was chosen because of its high response and sensitivity. The data were processed with Mass Hunter Workstation Data Acquisition Software. For full-scan MS analysis, the spectra were recorded in the range of 100–1700 *m*/*z*. The desolvation temperature was set to 350 °C at a flow rate of 700 L/h and a source temperature of 100 °C. The nebulizer pressure was set to 30 psig, the drying gas flow velocity to 12 L/min, and the flow velocity to 10 L/min. The capillary voltage was set at 3000 V and the nozzle voltage at 2500 V. For MS/MS automated and targeted analyses, the collision cell energy was set at 35 eV. The internal reference masses, including *m*/*z* 119.0363 (C_5_H_4_N_4_) and 966.0007 (C_19_H_20_F_24_N_3_O_8_P_3_) were used as a low flow of time-of-flight reference mixture and were introduced continuously with LC flow for accurate mass calibration.

### 3.4. UHPLC Method Validation

The UHPLC method was validated in terms of linearity, precision, accuracy, limit of detection (LOD), limit of quantification (LOQ), repeatability, and recovery. To construct calibrating curves, six standard stock solutions were mixed and diluted with acetonitrile to six appropriate concentrations. The linear curves were obtained by the peak area (*y*) versus standard concentration (*x*) of each analyte. The LODs and LOQs were determined by further dilution of standard solutions to a series of concentrations; these were injected into the UHPLC until the signal-to-noise ratios (S/N) were 3:1 and 10:1, respectively. The precision was determined by analyzing intra-day and inter-day variations. The samples were analyzed six times within the same day and on three consecutive days to determine the inter-day and intra-day precisions, respectively. The stability of the sample solution at room temperature was determined by replicated injection of sample solution at 0, 2, 4, 6, 12, and 24 h. Repeatability was investigated using six independent replicate experiments of the same sample. Recovery was obtained by adding a certain amount (low, medium, high) of the six analytes to 0.25 g of sample powder. All samples were extracted and analyzed as described in Section 3.3. Variances are expressed as relative standard deviations (RSD). 

### 3.5. Statistical Analyses 

Data were assessed by principal component analysis (PCA) using SIMCA 14.1 software (Umetrics, Umea, Sweden) and hierarchical cluster analysis (HCA) using SPSS (version 22.0, IBM, Somers, NY, USA). Heat map analysis was generated in MetaboAnalyst after data normalization. Statistical analysis was performed using one-way ANOVA followed by Tukey’s post-hoc tests and computed using GraphPad Prism software(GraphPad software, La Jolla, CA, USA). The criterion for statistical significance was *p* < 0.05. 

## 4. Conclusions

In the present study, quantification and qualification methods were established to evaluate the quality of NF subjected to different post-harvest processing methods. Six major bioactive constituents were simultaneously determined in all of the processed samples. The processing method had a considerable influence on the content of bioactive components in dried NF. The optimal drying method was hot air drying due to its less chemical transformation and high efficacy, and it should be used for the standardization of NF. Total ion current chromatography also indicated that there were significant changes in the chemical profiles of the processed samples. The influence of the post-harvest processing method on the pharmacological action of NF needs to be further investigated for better clinic application. The hot air-drying method should be used for standardization of NF based on the chemical studies. This study provides a basis for ongoing efforts to improve NF quality through the optimization of post-harvest processing.

## Figures and Tables

**Figure 1 molecules-24-03188-f001:**
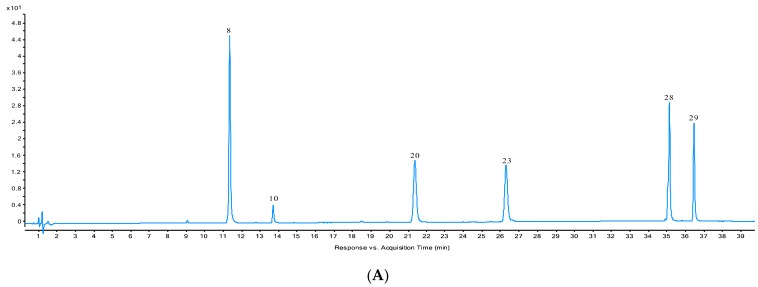
Typical DAD chromatograms of mixed reference compounds (**A**) and the average chromatograms of the sample of post-harvest processing methods (**B**).

**Figure 2 molecules-24-03188-f002:**
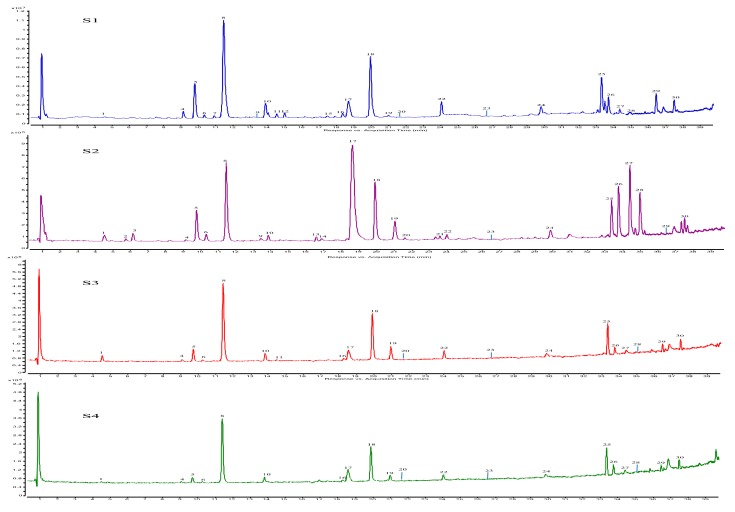
TIC chromatograms monitored in negative ion model fresh sample (S1) and shade drying (S2), Microwave drying(S3), Freeze drying (S4), Sunlight drying (S5), hot air drying(S6), Vacuum drying(S7), Far infrared ray drying (S8).

**Figure 3 molecules-24-03188-f003:**
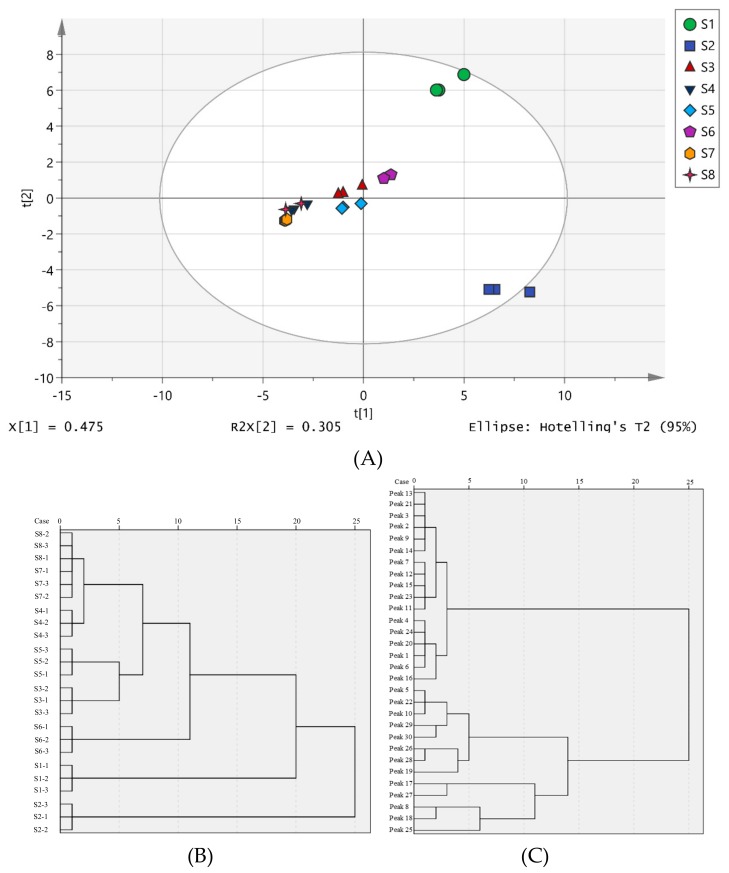
Multivariate statistical analysis of fresh sample and seven processed samples. (**A** and **B**) The PCA score plot and HCA plot of the LC-MS data set; (**C**) HCA plot of the LC-MS data based on chemical constituents. The compounds were identified either by MS2 spectra* or by authentic standards#. (S1–1~S1–3: Fresh sample, S2–1~S2–3: Shade drying, S3–1~S3–3: Microwave drying, S4–1~S4–3: Freeze drying, S5–1~S5–3: Sunlight drying, S6–1~S6–3: Oven drying, S7–1~S7–3: Vacuum drying, S8–1~S8–3: Far infrared ray drying) and different chemical constituents (peak 1–peak30).

**Figure 4 molecules-24-03188-f004:**
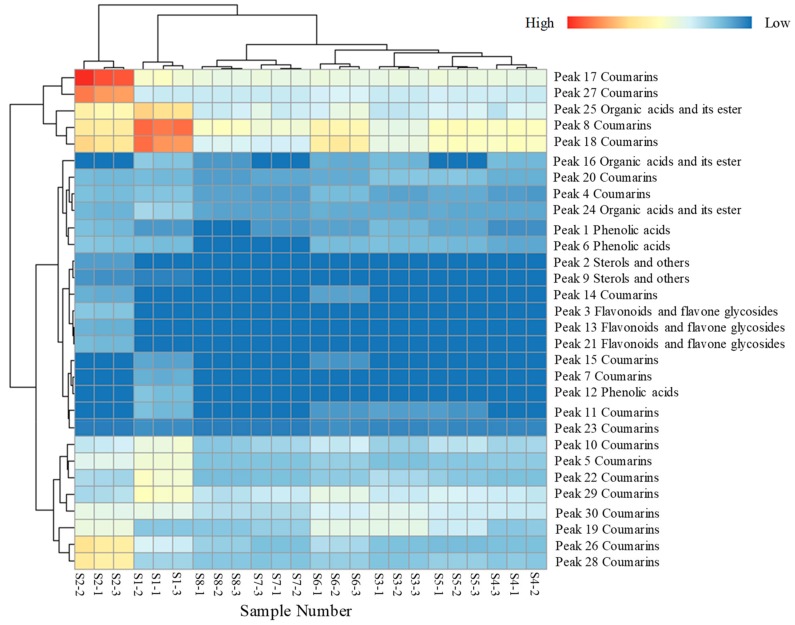
The heatmap analysis of annotated chemicals in fresh samples and seven processed samples by chemical categories.

**Table 1 molecules-24-03188-t001:** Validation of quantitative method by UHPLC-PDA.

Analyte	Y = ax + b		*r*	Range (μg/mL)	LOD (ng)	LOQ (ng)	Precision (*n* = 6) RSD	Stability (*n* = 6) RSD	Repeatability (*n* = 6) RSD	Recovery (*n* = 6) RSD
	a	b								
nodakenin	2.95 × 10^6^	−1127	0.9996	1.18–472	26	78	1.09%	1.46%	1.23%	2.56%
psoralen	2.43 × 10^7^	821	0.9999	0.44–176	15	44	1.11%	2.25%	1.79%	1.75%
bergapten	5.48 × 10^7^	873	0.9999	0.35–140	10	28	1.12%	0.95%	1.51%	0.74%
notopterol	2.97 × 10^7^	240	0.9999	0.58–234	19	58	0.57%	2.41%	1.03%	1.43%
imperatorin	2.88 × 107	725	0.9999	0.30–120	16	48	0.74%	2.87%	2.03%	2.06%
isoimperatorin	1.42 × 10^7^	6764	0.9999	0.20–80	9	26	0.42%	0.95%	1.42%	0.99%

**Table 2 molecules-24-03188-t002:** The mean determination of six bioactive components in fresh *N. franchetii*, slices dried with different methods (mg/g, *n* = 3).

No.	Drying Method	Nodakenin	Psoralen	Bergapten	Notopterol	Imperatorin	Isoimperatorin	Summary	Reduction Rate (%)
S1	Fresh sample	24.31	0.17	0.30	0.13	0.57	9.79	35.27	-
S2	Shade drying	14.84	0.08	0.31	0.06	5.17	3.06	23.52	33.31
S3	Microwave drying	7.18	0.04	0.40	0.07	0.21	5.00	12.90	63.44
S4	Freeze drying	10.4	0.05	0.22	0.07	0.28	4.92	15.94	54.80
S5	Sunlight drying	10.94	0.07	0.39	0.08	0.39	5.55	17.42	50.61
S6	Hot air drying	14.22	0.08	0.21	0.11	0.53	8.99	24.14	31.55
S7	Vacuum drying	8.75	0.05	0.16	0.06	0.19	5.08	14.29	59.47
S8	Far infrared ray drying	10.12	0.03	0.10	0.06	0.22	4.27	14.80	58.04

**Table 3 molecules-24-03188-t003:** The compounds identified in the TIC chromatogram.

Peak No.	Retention (min)	Molecular Formula	Quasi-molecular[M-H] = [M + Cl/COOH]^−^(error, ppm)	Quasi-Molecular[M + H/Na]^+^(error, ppm)	MS/MS Fragments Ions	Identification
**1**	4.495	C_16_H_18_O_9_	353.0878(0.02)		191.0466,173.0452,135.0427	Chlorogenic acid [24]
**2**	5.754	C_29_H_50_O		415.3959(−5.92)	399.1085,255.0318,211.9974	β-Sitosterol [25]
**3**	6.11	C_28_H_32_O_15_		609.1819(−0.77)	463.1250,301.0724,203.0353	Diosmin [25]
**4**	9.17	C_9_H_6_O_3_	161.0244(−0.08)		133.0297,119.0194,106.2345	Umbelliferone [26]
**5**	9.779	C_20_H_24_O_9_		409.1482(2.71)	246.9364,228.8693,174.5889	Isomer of nodakenin [27]
**6**	10.313	C_10_H_10_O_4_	193.0506(1.20)		178.0160,134.0372,105.0353	Ferulic acid [28]
**7**	10.941	C_13_H_10_O_5_	245.0479(−9.6)		227.0690,211.0387,159.0454	Isopimpinellin [29]
**8**	11.463	C_20_H_24_O_9_		409.1482(2.71)	246.9364,228.8695,174.5887	Nodakenin [27]
**9**	13.416	C_21_H_32_O_2_		317.2474(0.34)	183.0106,119.0331,102.0110	Pregnenolone [25]
**10**	13.847	C_11_H_6_O_3_		187.0402(−6.57)	143.0504,131.0500,115.0546	Psoralen [25]
**11**	14.505	C_20_H_24_O_10_		447.1233(6.41)	241.0875,179.0853,127.0393	Decuroside V [30]
**12**	14.978	C_25_H_24_O_12_		517.1314(5.13)	355.1988,200.7070,156.4883	3,5-Dicaffeoylquinic acid [31]
**13**	16.581	C_21_H_24_O_11_		453.1364(6.04)	322.2452,283.1529,208.8737	(−)-Catechin-7-*O*-glucoside [32]
**14**	16.894	C_16_H_16_O_4_	271.0970(−0.43)		225.2215,137.0230,106.0417	*p*-Hydroxyphenethyl anisate [25]
**15**	17.442	C_26_H_26_O_12_	529.1315(6.9)		363.1776,247.1300,159.0230	1-Caffeoyl-5-feruloylquinic acid [33]
**16**	18.314	C_16_H_16_O_6_		305.0991(9.39)	202.7124,174.58312,158.5002	Oxypeucedanin hydrate [34]
**17**	18.761	C_11_H_6_O_4_	203.0354(−7.46)		159.0448,147.0450,131.0502	Bergaptol [14]
**18**	19.943	C_30_H_32_O_12_		585.1998(−5.38)	405.1338,247.0979,177.0560	6′-O-*trans*-feruloyl nodakenin [27]
**19**	20.933	C_16_H_14_O_4_		271.0959(2.16)	201.0558,173.0608,145.0658	Isomer of imperatorin [35]
**20**	21.614	C_12_H_8_O_4_	261.0424(−7.43)		201.7049,176.6158,145.4164	Bergapten [35]
**21**	23.623	C_16_H_12_O_6_	299.0566(−1.63)		284.0314,256.0362,183.0440	Diosmetin [14]
**22**	24.016	C_21_H_22_O_4_	337.1440(1.58)		201.0172,173.0205,109.0287	Anhydronotoptol [7]
**23**	26.571	C_21_H_22_O_5_		355.1566(−7.32)	172.5619,216.7848,272.8885	Notopterol [35]
**24**	29.892	C_16_H_14_O_5_		287.0936(−7.66)	203.0355,175.0402,147.0450	Oxypeucedanin [7]
**25**	33.391	C_18_H_18_O_5_	349.0858(−2.79)		149.0610,134.0364,117.0335	*p*-Hydroxyphenethyl ferulate [31]
**26**	33.725	C_18_H_18_O_4_	297.1137(−1.57)		183.0132,160.8449,136.9432	Phenethyl ferulate [35]
**27**	34.376	C_20_H_26_O_4_	329.1756(0.71)		314.1504,177.0181,133.0289	Bornyl ferulate [35]
**28**	34.948	C_16_H_14_O_4_		271.0959(2.16)	201.0557,173.0609,145.0658	Imperatorin [35]
**29**	36.501	C_16_H_14_O_4_		271.0959(2.16)	201.0559,173.0608,145.0658	Isoimperatorin [35]
**30**	37.301	C_19_H_22_O_3_		299.1628(4.58)	175.0400,145.0295,119.0498	7-Geranyloxycoumarin [35]

The compounds **8**, **10**, **20**, **23**, **28** and **29** were confirmed by comparison with chemical standards.

**Table 4 molecules-24-03188-t004:** Relative fold change of the main compounds in eight samples.

Peak No.	Retention (min)	Fold Change							
		Fresh Sample	Shade Drying	Microwave Drying	Freeze Drying	Sun Drying	Hot Air Drying	Vacuum Drying	Far infrared Ray Drying
**1**	4.495	1.00	3.47	3.31	0.29	1.85	1.76	0.89	0.00
**2**	5.754	0.00	1.00	0.00	0.00	0.00	0.00	0.00	0.00
**3**	6.110	0.00	1.00	0.00	0.00	0.00	0.00	0.00	0.00
**4**	9.170	1.00	0.88	0.42	0.28	0.52	0.84	0.30	0.39
**5**	9.779	1.00	0.76	0.06	0.16	0.13	0.20	0.08	0.07
**6**	10.313	1.00	1.13	0.98	0.41	0.87	0.95	0.00	0.00
**7**	10.941	1.00	0.00	0.00	0.00	0.00	0.00	0.00	0.00
**8**	11.463	1.00	0.61	0.30	0.43	0.45	0.58	0.36	0.42
**9**	13.416	1.00	1.90	0.00	0.00	0.00	0.00	0.00	0.00
**10**	13.847	1.00	0.47	0.24	0.29	0.41	0.47	0.29	0.18
**11**	14.505	1.00	0.00	0.38	0.00	0.77	0.81	0.00	0.00
**12**	14.978	1.00	0.00	0.00	0.00	0.00	0.00	0.00	0.00
**13**	16.581	0.00	1.00	0.00	0.00	0.00	0.00	0.00	0.00
**14**	16.894	0.00	1.00	0.00	0.00	0.00	1.09	0.00	0.00
**15**	17.442	1.00	0.00	0.00	0.00	0.00	0.88	0.00	0.00
**16**	18.314	1.00	0.00	0.90	0.62	0.00	0.58	0.00	0.39
**17**	18.761	1.00	6.17	0.27	0.29	0.41	0.39	0.33	0.32
**18**	19.943	1.00	0.73	0.19	0.33	0.33	0.48	0.08	0.12
**19**	20.933	1.00	9.38	8.62	1.19	4.59	6.94	1.61	1.07
**20**	21.614	1.00	1.03	1.33	0.73	1.30	0.67	0.53	0.33
**21**	23.623	0.00	3.66	0.00	0.00	0.00	0.00	0.00	0.00
**22**	24.016	1.00	0.31	0.34	0.12	0.19	0.21	0.13	0.09
**23**	26.571	1.00	0.46	0.54	0.54	0.62	0.77	0.46	0.46
**24**	29.892	1.00	0.48	0.34	0.30	0.29	0.38	0.26	0.27
**25**	33.391	1.00	0.77	0.26	0.36	0.34	0.47	0.29	0.30
**26**	33.725	1.00	2.92	0.25	0.21	0.19	1.34	0.20	0.85
**27**	34.376	1.00	25.62	0.64	0.93	1.20	2.41	0.89	0.83
**28**	34.948	1.00	8.72	0.36	0.50	0.69	0.92	0.34	0.38
**29**	36.501	1.00	0.38	0.51	0.50	0.57	0.82	0.52	0.44
**30**	37.301	1.00	1.04	0.93	0.58	0.61	0.72	0.23	0.34

**Table 5 molecules-24-03188-t005:** The drying duration and final moisture of *N. franchetii* dried under the different methods.

No.	Drying Methods	Drying Parameters	Duration	Moisture (%)
		Temp. or Power		
S2	Shade drying	22–25 °C	10 d	9.35
S3	Microwave drying	2450 MHz	30 min	8.75
S4	Freeze drying	−35 °C	8 h	9.28
S5	Sunlight drying	22–30 °C	4 d	10.25
S6	Hot air drying	50 °C	2 d	10.35
S7	Vacuum drying	50 °C	4 h	7.82
S8	Far infrared ray drying	80–85 °C, 1125 W	3h	8.54

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
