# Peer review of "Effect of Different Post-Harvest Processing Methods on the Chemical Constituents of *Notopterygium franchetii* by an UHPLC-QTOF-MS-MS Metabolomics Approach"

_molecules, 2019, doi:10.3390/molecules24173188_

Round 1

Reviewer 1 Report

Introduction:

“Among them, coumarins, especially isoimperatorin, notopterol, and bergapten, are well known to be the major bioactive compounds, and contribute to NF’s anti-inflammatory, analgesic, and anti-cancer activities[8-9].”

Could you add references with deal with coumarins biological activities in Notopterygium spp?

Results (3.3 Quantification of six analytes in fresh NF and NF dried by different methods)

-“The content in far infrared radiation dried samples decreased by 63.44%, which was the most significant change in all samples. This interesting phenomenon suggests that far infrared radiation drying destroys some of the chemical constituents”.

Table 3 reported that the far infrared ray drying has a reduction rate of  58.04% and not 63.44%.

-“Obviously, the durations required for vacuum and hot air drying were much shorter than those of sunlight and shade drying, which may lead to greater degradation of all six components”

According to the Table 1, hot air drying was not the shorter. Furthermore, I suggest to write  again the sentences because it seems that a short duration leads to a greater degradation of all six components.

“In summary, shade and sunlight drying were superior to far infrared radiation and vacuum drying.”

What does “Superior” means?

“Obviously, shorter drying durations can preserve better for the bioactive constituents….. However, due to its long drying duration, hot air drying in the oven resulted in a similar chemical composition to that of fresh”. It is not clear

In my opinion, this paragraph should be written more clearly

3.5 Chemical trasformation of NF during drying

"The increase in compound 17 after shade drying was 10.0-fold..."

What does "sensitive" to shade drying means?

Conclusions

Could you explain better the reason for which hot air drying method is optimal drying method?

Reviewer 2 Report

the manscript is the cientific interest and is very well written

Author Response

We have consulted native English speakers for paper revision and attached here the statement of editing.

Reviewer 3 Report

The paper is very well written, well researched, experiments carried put scientifically. All the methods are presented clearly. Results have been analyzed and presented well.

Author Response

The paper had been revised by a native English speaker. The statement of editing was attached here.

Reviewer 4 Report

This work is a well-done research in the field of analytical chemistry. From my point of view, it is suitable for publication in Molecules after major revision.

Authors have compared extraction systems, nevertheless they did not include in the manuscript an optimization for each system. How did authors select extraction parameters?  it has been described the type of solvent, but for example optimization of microwave or freeze drying (first and second freeze drying parameters) has not been described and has to be included in the new  version. In addition, more details about technical parameters could provide useful information  in order to carry out the extraction in other laboratory.

Concerning optimization of HPLC analysis, authors have reported the optimization of the chromatographic separation using different columns, mobile phase, temperature and gradients. This part has to be rewritten for a better understanding. Authors selected a column based on the best chromatographic separation under a specific conditions. Since the stationary phases are different more than one conditions should be tested before to choose a column. Author should justify the selection of the column. 

Concerning detector, is DAD or PDA? Optimization of wavelength? Or selection?

Author Response

Please see the attachment. English language and some minor spell has been checked by a native English speaker.

Round 2

Reviewer 4 Report

The new version is suitable to be published in Molecules